# Analysis and Evaluation of Energy Consumption and Carbon Emission Levels of Products Produced by Different Kinds of Equipment Based on Green Development Concept

**Yongmao Xiao [1,2], Renqing Zhao [3,\*], Wei Yan [4] and Xiaoyong Zhu [5]**

1   School of Computer and Information, Qiannan Normal University for Nationalities, Duyun 558000, China; xym198302@163.com
2   Key Laboratory of Complex Systems and Intelligent Optimization of Qiannan, Duyun 558000, China
3   School of Maxism, Anhui Science and Technology University, Bengbu 233030, China
4   Academy of Green Manufacturing Engineering, Wuhan University of Science and Technology, Wuhan 430081, China; yanwei81@wust.edu.cn
5   School of Economics & Management, Shaoyang University, Shaoyang 422099, China; zhuxysyu@126.com
\*   Correspondence: zhaojia7933@163.com

**Abstract:** Energy consumption and carbon emission levels in the production process constitute an important basis for the selection of production equipment. The energy consumption and carbon emission levels of the same product produced by different kinds equipment vary greatly from one tool to another. Unfortunately, traditional modes of selection only give qualitative results, so that it is difficult to provide a quantitative reference to enable enterprises to choose appropriate modes of production in the context of the green development concept (GDC). In order to solve this problem, a calculation method for multiple energy consumption and carbon-emission objectives for commodity production is proposed. The focus of this paper is to research the difference between the energy consumption and carbon emission levels of the same product produced by different kinds of equipment. The energy consumption and carbon emissions of different kinds of equipment can be calculated by gray wolf algorithm. The results show that the proposed method can calculate the optimal values of energy consumption and carbon emissions in the same kinds of products produced by different equipment, which can provide assistance for enterprises in choosing the production equipment that best conforms to the green development concept.

**Keywords:** green development concept; product production process; energy consumption; carbon emission; level analysis and evaluation

## 1. Introduction

In recent years, the rapid development of the product manufacturing industry has seriously overdrawn resources and damaged the environment. Global environmental problems, such as the greenhouse effect, acid rain, haze, ozone consumption, air pollution, water-source pollution, land pollution, river closure, land desertification, and soil erosion, threaten the survival of mankind [1–3]. Traditional industries characterized as high-input, high-consumption, high-pollution, low-quality, low-benefit, low-output, and high-pollution, no longer meet social development needs. The achievement of harmonious co-existence for humans and nature has become an urgent problem that must be solved in order to maintain socially sustainable development. Green development is a new concept proposed in response to the resource bottleneck and environmental problems created by traditional modes of development and economic rationalism. The central idea of green development is an ecological approach, which is to address the problem of the hostile relationship between human social activities and the natural environment. Green manufacturing is a necessary support for green development. Commodity production is an

important basis for the survival of manufacturing enterprises, and a significant source of resource consumption and environmental pollution. Analysis and evaluation of resource consumption and carbon emissions in the production process have become an effective method for promoting green development in the manufacturing industry [4–6].

In their research on the selection of methods of commodity-production process planning, Zhang et al. [7] established a decision-making model of green process planning verified by the analytic hierarchy process (AHP). Cheng et al. [8] established a multi-process low-carbon manufacturing decision-making model, and verified it. Wu et al. [9] established a product process-plan green analysis technology based on the decision maker's subjective preference and feedback of decision information. The method was illustrated and verified by the green analysis of an automotive component process plan. Li et al. [10] put forward a method of product process-planning scheme evaluation based on extension analysis and DS theory. The effectiveness of the proposed method was verified by the application of a case study on the process planning of the rear door frame of an automobile. Guo et al. [11] proposed the precision machine tool assembly adjustment-process decision method based on the accuracy of precision machine tools and the error state optimal estimation. An et al. [12] presented a new method of multiple attribute decision-making based on rough fuzzy number. Zhou et al. [13] proposed an interval number approach to ideal ranking method decision model for green process planning. Fuzzy analytic hierarchy process (FAHP) and TOPSIS were used to solve the model. Wang et al. [14] proposed a motivation and risk-assessment decision and planning method; the method can effectively advocate real-time decision-driving behavior according to the current environment. Jiskani et al. [15] proposed and tested an indicator framework for analyzing and prioritizing the identified indicators in order to render technical assistance for the implementation of green and climate-smart mining. Xu [16] constructed an evaluation index system to objectively and accurately assess the green innovation capability of manufacturing enterprises. The feasibility of the model and the stability of the evaluation results were verified. Wang et al. [17] used a hybrid multiple-criteria decision-making method for additive-manufacturing process selection. These studies only put forward qualitative methods of product planning selection, and did not consider the dynamic relationship between production process and the selection process, so it is difficult to quantify the impact of each process-plan selection on the decision-making goals.

For the optimization of machining parameters, Yang et al. [18] established a mapping model of energy consumption and main process parameters for a cold-rolling continuous annealing line, and verified the solution by NSGA-II. Liu et al. [19] put forward an integrated optimization model of cutting parameters and scheduling, aiming at minimizing the carbon emissions and completion time of the manufacturing process, and used the improved multi-objective gravity search algorithm to verify it. Zhang et al. [20] established a multi-objective optimization model with carbon emissions and noise as optimization objectives, and solved the model by using the adaptive niche genetic algorithm. Zhan et al. [21] established a multi-objective optimization model, and proposed an improved non dominated sorting gravity search algorithm to solve the multi-objective model. Li et al. [22] constructed a laser-welding energy-consumption and welding-quality mode and tested it. Tian et al. [23] established a multi-objective cutting parameter optimization model and tested it using modified NSGA-II algorithm. Zhou et al. [24] set up an optimization model and used improved NSGA-II with non-cooperative game theory to verify it. Zhao et al. [25] proposed a dynamic cutting parameter optimization method based on digital twin; this method can dynamically find the optimal cutting parameters according to real-time sensor data of machining conditions. Xiao et al. [26] established a multi-objective optimization model, and proposed a combined optimization algorithm based on particle swarm optimization and NSGA-II to solve the model. In the above study, the production process parameters were optimized, and quantitative analysis was used to compare and verify the data. However, these studies only researched types of device or equipment, without considering the problem of processing the same product with different kinds of equipment.

The production of industrial commodities is an important basis for the sustainable operation of society [27,28]. The same product can be processed with different equipment, and the energy consumption and carbon emission values of equipment production processes are quite different from each other. In the actual process, the equipment selection usually relies on qualitative analysis rather than quantitative calculation, which cannot provide the reference for selecting the suitable equipment under green-development conditions. In this paper, a unified energy consumption and carbon emission calculation model for multiple equipment is established, and energy consumption and carbon emission objectives of the same product processed with different equipment production processes are solved by using NSGA-II algorithm to provide suggestions for the selection of production methods.

The framework of the article is shown in Figure 1. The first chapter introduces the literature related to the process-selection evaluation of the production process and the optimization of single equipment processing parameters, and puts forward the problems existing in the current research. Then, the problem of analyzing and evaluating the energy consumption and carbon emission levels of a product in the processing of multiple types of equipment is proposed. In the second chapter, based on the green development concept, a framework model of production-process analysis and evaluation is established. The third chapter builds a unified calculation model for multiple types of equipment. In Section 4, the case study is conducted to analyze the processing characteristics of a given product made with multiple types of equipment, optimize the production process with gray wolf optimization algorithm, and analyze and evaluate the energy consumption and carbon emission levels of different processing equipment. The fifth chapter is the summary of the article.

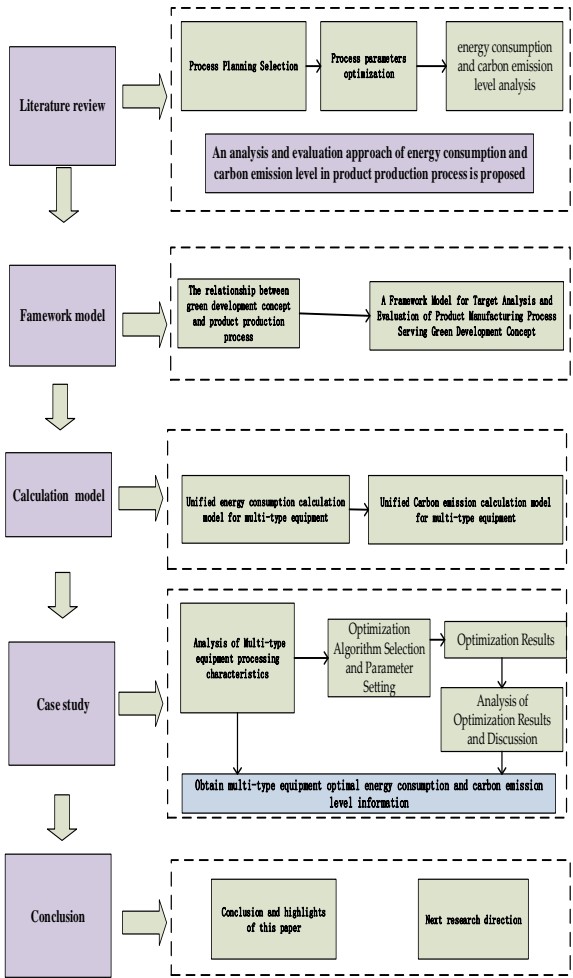

**Figure 1.** Overall research framework.

## 2. Analysis and Evaluation Framework Model of Production Process Serving the Concept of Green Development

The production process is an important source of energy consumption and environmental pollution. It is important to guide the production process according to the principles of green development. Macroscopically, the logical relationship between the product production process and green development is analyzed. At the micro level, a framework model of product process analysis and evaluation based on service GDC is established along the main line of material selection, product process analysis, production target selection, target level analysis, and evaluation process. The framework model aims to discover the weak link of green transformation; upgrade, design, and evaluate the level of green development in commodity production; and guide the green transformation and upgrade of enterprises.

### 2.1. The Relationship between GDC and Product Production Process

The core of the GDC is "ecology". Global economic development and industrialization have generated ecological pressures, such as smog, water quality deterioration, and other ecological problems [29–31]. The concept of green development proposes to guide the transition of the economic development model from consumption type to economy type, from pollution type to clean type, and from high-carbon type to low-carbon cycle type. Its implementation would also change the development model, from pursuing the development of the economy only to the pursuit of all-round development focusing on "green". The formation and implementation of the concept of green development is the basis of social sustainable development. GDC is the guiding concept of developing green industry, which is an important basis for realizing the concept of green development. One by one enterprises constitute industry, while the implementation of the concept of green development of enterprises constitute a green industry. The important task of an enterprise is to produce products. The profit of an enterprise is composed of product profits. The product manufacturing process is an important source of resource consumption and environmental damage. Resource conservation and environmental protection awareness of product production aim to promote the realization of green development [32,33]. The relationship between the green development concept and the product production process is shown in Figure 2.

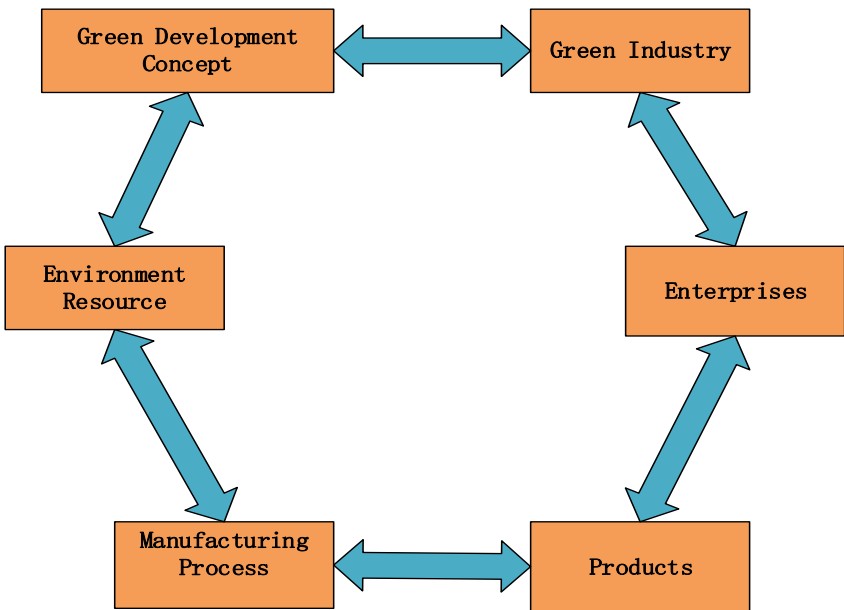

**Figure 2.** Green development concept and product production process.

### 2.2. A Framework Model for Objective Analysis and Evaluation of Product Manufacturing Process Serving GDC

The frame model of objective–level analysis and evaluation of the product manufacturing process serving the concept of green development is the practice of applying the integrated idea to the optimization of the production process. According to GDC, based on the equipment and processes available in the manufacturing process, with the target system of productivity, quality, cost, resource consumption, and environmental impact as the core, the frame model will integrate throughout the overall production process optimization and the entire process [34–36]. The framework model is an integration of optional equipment, process objective system, and optimization calculation. The objective analysis and evaluation of the production process adopts modular analysis, and its block diagram model is shown in Figure 3.

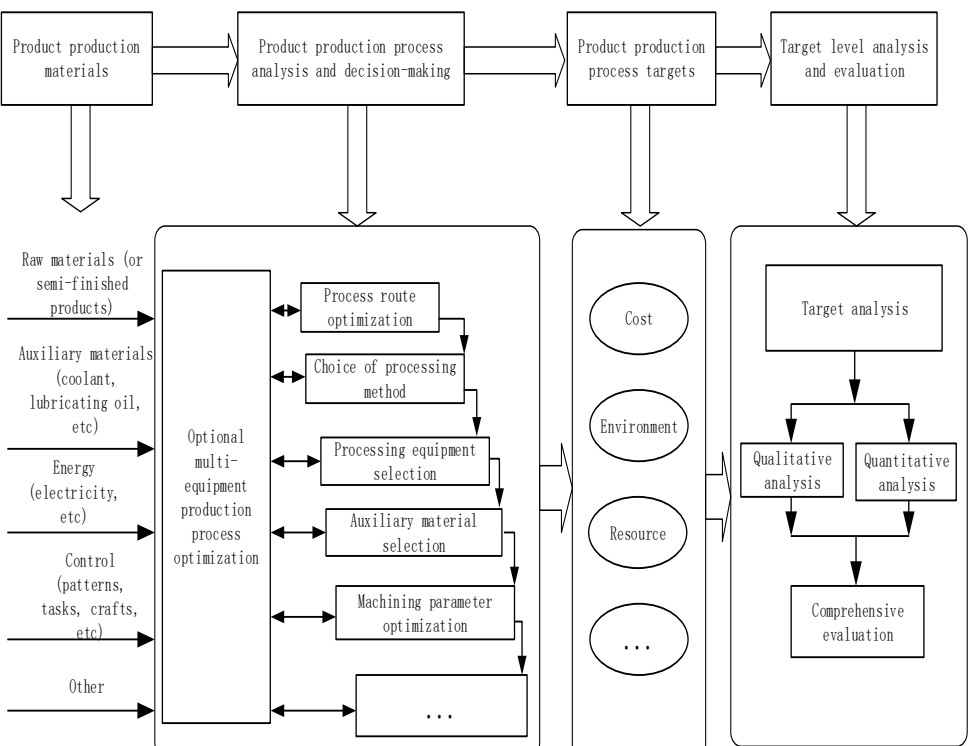

**Figure 3.** Analysis and evaluation framework model of the product manufacturing process serving the concept of green development.

## 3. Unified Calculation Model of Green Development Objective in Different Equipment Production Process

There are many kinds of equipment that can be used in the process of product production. It is necessary to choose suitable equipment to establish a model with a set of unified energy consumption and carbon emission target functions.

### 3.1. Energy Objective

The establishment of a unified cutting model is an important basis for attaining green development objectives and for the analysis of different equipment production processes. The energy consumption calculation process is shown in Figure 4. This method includes the following steps [37–41].

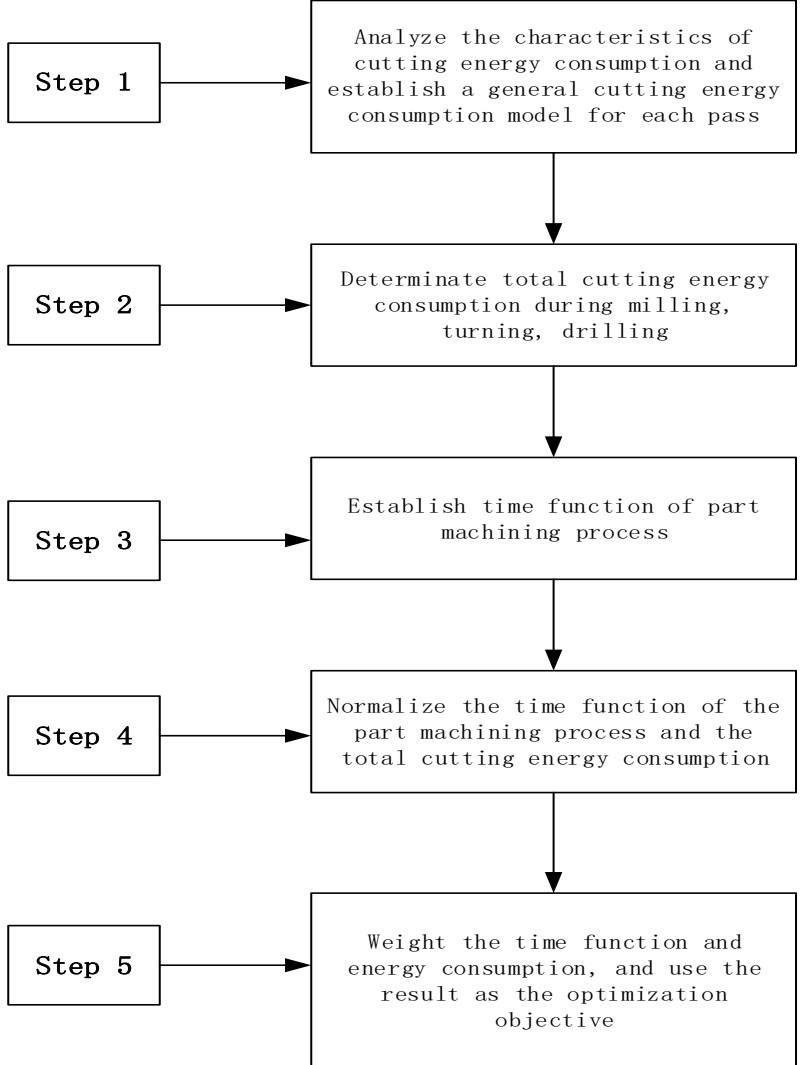

**Figure 4.** Energy consumption calculation process.

(1)   Establish the cutting energy consumption model.

$$E = SEC \cdot V + P_{airi} \cdot \Delta t_{airi} \tag{1}$$

A general model of cutting energy consumption for each feed:

$$E_i = SEC_i \cdot V_i + P_{airi} \cdot \Delta t_{airi} \tag{2}$$

where $E$ is the cutting energy consumption; $V$ is the volume of material removed; $P_{airi}$ is the empty cutting power; $\Delta t_{airi}$ is the empty cutting process time; $i$ is the serial number of the feed, and $E_i, SEC_i, P_{airi}, t_{airi}$ corresponding to the $i$ feed of $E, SEC, P_{air}, t_{air}$.

$$SEC = \frac{P_{normal}}{MRR} = k_1 \frac{n}{MRR} + k_2 MRR^{k_3} + k_4 \frac{1}{MRR} \tag{3}$$

where $P_{normal}$ is the cutting stage power; $MRR$ is the material re removal rate; $k_1$ is the constant coefficient obtained by the experiment; $n$ is the spindle speed; $k_2$ is the power constant coefficient related to the type of the machine tool during the cutting process; $k_3$ is a constant related to the type of the machine tool during the cutting process; $k_4 = P_{standby} + P_{fluid} + a$ is the constant coefficient in the cutting process; $P_{standby}$ is

the machine tool standby power; $P_{fluid}$ is the cutting fluid consumption of power, and $a$ is the experimentally obtained power constant.

$$P_{air} = P_{standby} + P_{fluid} + k_1 n + a + k_5 f + b \tag{4}$$

where $f$ is feed rate, $k_5$ and $b$ are the power constant coefficient of the feed motor.

(2) Calculate the total cutting energy consumption.

$$E_{ZO} = \sum_{i=1}^{m} E_i = \sum_{i=1}^{m} (SEC_i \cdot V_i + P_{airi} \cdot \Delta t_{airi})$$
$$= \sum_{i=1}^{m} \left[ \left( k_1 \frac{n_1}{MRR_i} + k_4 \frac{1}{MRR_i} + k_2 MRR_i^{k_3} \right) \cdot V_i + (k_1 n_1 + k_5 f_i + c) \cdot \Delta t_{airi} \right] \tag{5}$$

among, $E_{ZO}$ is the total energy consumption of cutting; $m$ is the number of walking knives in the processing process; $V$ is the volume of material removed; $P_{airi}$ is empty cutting power; $\Delta t_{airi}$ is the air cutting process time; superscript $i$ is the serial number of the feed; $E_i$, $SEC_i$, $P_{airi}$, $t_{airi}$ correspond to the feed $i$; $MRR$ is the material removal rate; $k_1$ is the constant coefficient obtained by the experiment; $n$ is spindle speed; $k_2$ is the power constant coefficient related to the type of the machine tool during the cutting process; $k_3$ is a constant related to the type of the machine tool during the cutting process; $k_4$ is the constant coefficient in the cutting process; and $k_5$ is the power constant coefficient of the feed motor.

(3) Establish a function of time for the part machining process.

$$t_w = \frac{\pi dV}{1000 V_c f_t Z a_p a_e} + \frac{t_{ct} \pi dV V_c^{x-1} a_p^{y-1} f_t^{u-1} a_e^{w-1} Z^{q-1}}{1000 C_T} + t_{ot} \tag{6}$$

In the formula, $t_w$ is the function of time for the part machining process; d is the knife diameter; $Z$ is the number of teeth; $t_{ct}$ is the time taken to change the knife at once; $C_T$ is the coefficient, related to the workpiece material, cutting conditions, and the tool itself; $x, y, u, w, q$ are the index, which represents the influence of each milling amount on tool durability; $t_{ot}$ is the auxiliary time outside of the knife change process; $V_c$ is cutting speed; $f_t$ is cutting speed; $a_p$ is the axial cutting depth; and $a_e$ is the radial cutting depth.

(4) Normalize the time function and total energy consumption. The process of normalized processing time function and total energy consumption is as follows:

$$t_w^* = \frac{t_w(V_c, f_t, a_p, a_e) - t_{wmin}}{t_{wmax} - t_{wmin}} \tag{7}$$

In the formula, $t_w^*$ is the function of time for the normalized part machining process; and $t_{wmax}$ and $t_{wmin}$ are the minimum and maximum values optimized only for processing time, respectively.

$$E_{ZO}^* = \frac{E_{ZO}(V_c, f_t, a_p, a_e) - E_{ZOmin}}{E_{ZOmax} - E_{ZOmin}} \tag{8}$$

In the formula, $E_{ZO}^*$ is the total cutting energy consumption after normalization treatment; $E_{ZOmax}$ and $E_{ZOmin}$ are the minimum and maximum optimization values of processing energy consumption.

(5) Calculate the expression of energy consumption optimal objective. The processing parameter optimization method based on the general cutting energy consumption model is as follows:

$$\min F(V_c, f_t, a_p, a_e) = \min(w_1 t_w^* + w_2 E_{ZO}^*)$$
$$= \min(w_1 \frac{t_w(V_c, f_t, a_p, a_e) - t_{wmin}}{t_{wmax} - t_{wmin}} + w_2 \frac{E_{ZO}(V_c, f_t, a_p, a_e) - E_{ZOmin}}{E_{ZOmax} - E_{ZOmin}}) \tag{9}$$

In the formula, $w_1$ and $w_2$ are the weight coefficients; $w_1 + w_2 = 1$.

### 3.2. Carbon Emission Objective

The carbon emission in the production process mainly includes the carbon emissions caused by the consumption of raw materials, electric energy, the post-processing of chips in the production process, and ancillary materials in the production process [42–49]. The emission of carbon in the production process is shown in Figure 5.

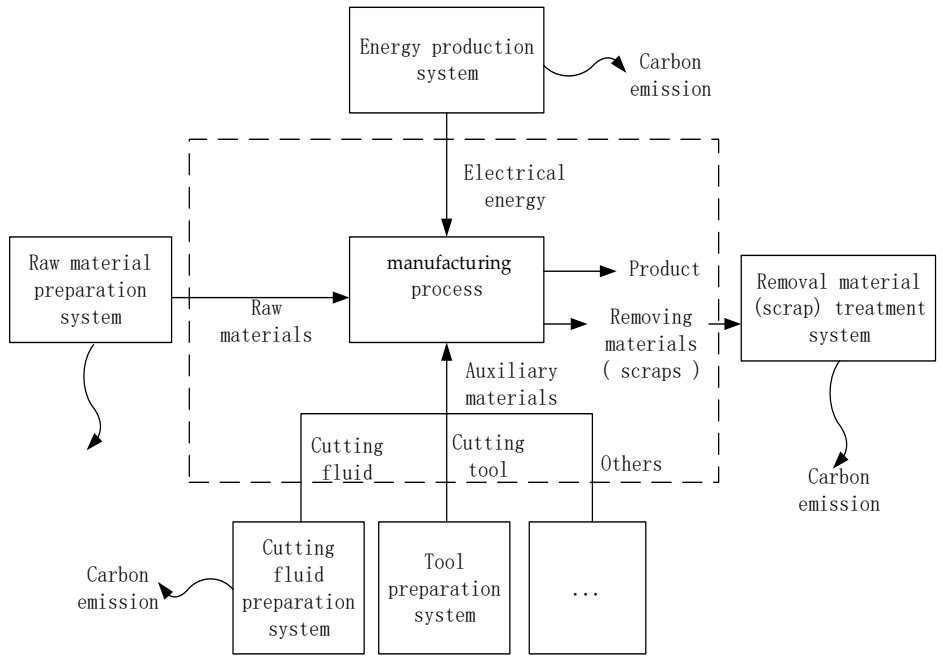

**Figure 5.** Chart of carbon emission from manufacturing process.

The emission of carbon in the production process can be expressed as:

$$C_p = C_e + C_t + C_c \tag{10}$$

(1) The electric energy carbon emission. In the process of CNC machining, a large amount of electric energy needs to be consumed. The carbon emitted due to electrical energy consumption $C_e$ during the NC machining process is calculated as follows:

$$C_e = F_e E_{ZO} \tag{11}$$

In the formula, $F_e$ represents the carbon emission factor of the electrical energy (KGCO$_2$/kWh)); $E_{ZO}$ represents the electrical energy consumption of the process, which is shown in Equation (5); and $F_e$ is closely related to the composition of the power grid. Different power grids have different carbon emission factors. According to [46], 0.6747 is used as the carbon emission factor.

(2) Carbon emissions from tool use. In the process of machining, the carbon emission caused directly by the cutting tool is small, and is mainly due to the combination of the cutting tool preparation process and the use of the cutting tool. Therefore, the carbon emission of the tool is calculated by the time-standard conversion to the process distribution method in the tool life cycle. The specific calculation method is as follows:

$$C_t = \frac{t_m}{T_t} F_t W_t \tag{12}$$

where $F_t$ is the carbon emission factor for the tool, and $W_t$ is the quality coefficient of the tool. To determine the carbon emission factor of cutting tool $F_t$, we need to

know the process of cutting tool preparation and the energy consumption in the process of the cutting tool preparation. According to [47], the carbon emission factor in the process of cutting tool preparation is 29.6 kgCO$_2$/kg. Tool life $T_t$ refers to the cutting time experienced by a new tool until scrapped, which may include multiple regrinding (regrinding times expressed by $N$) time; thus, tool life is equal to the product of tool life $T$ and ($N + 1$),

$$T_t = (N+1)T \tag{13}$$

(3) Cutting fluid uses carbon emissions. The calculation of the carbon emissions of cutting fluid mainly takes into account the water-based cutting fluid in the NC machining process. Ascertaining carbon emission of cutting fluid mainly considers the carbon emission of pure mineral oil preparation $C_o$ and the carbon emission of cutting fluid disposal $C_w$. The calculation of carbon emissions from cutting fluids is converted to the machining process by time standards during its replacement cycle. The carbon emissions from the beechwood cutting fluid are calculated as follows:

$$C_c = \frac{T_p}{T_c}(C_o + C_w) \tag{14}$$

$$C_o = F_o(C_C + A_C) \tag{15}$$

$$C_w = F_w[(C_C + A_C)/\delta] \tag{16}$$

In the formula, $F_o$ is the pure mineral oil emission factor; $F_w$ is the waste cutting fluid treatment carbon emission factor; $C_C$ is the initial cutting oil consumption, and $A_C$ is the additional cutting oil consumption. $\delta$ is cutting fluid concentration; $T_c$ is the cutting fluid replacement cycle, and $T_p$ is processing time. The carbon emission factor for cutting fluid is divided into two parts, including the preparation of pure mineral oil required for the configuration of cutting fluids $F_o$ and the carbon emission factor of waste cutting fluid treatment $F_w$ The formula for calculating l $F_o$ is as follows:

$$F_o = E_{Eo}E_{Co} \times \frac{44}{12} \tag{17}$$

In the formula, $E_{Eo}$ is the intrinsic energy of the mineral oil (GJ/l), and $E_{Co}$ is the default carbon content of the mineral oil (kgc/GJ). According to [47], $F_o$ can be calculated as the value 2.85 kgCO$_2$/L. The carbon emission factor of waste cutting fluid treatment $F_w$ is 0.2 kgCO$_2$/L.

### 3.3. Constrains

The value of two objective functions is limited by the cutting parameters and quality requirements for parts, which can only be taken within the range, as shown in Equations (18)–(21).

(1) Cutting depth constraint. The cutting depth $a$ must between the maximum cutting depth $a_{max}$ and the minimum cutting depth $a_{min}$.

$$a_{min} \leq a \leq a_{max} \tag{18}$$

(2) Feed constraint. The feed must be between the minimum $f_{min}$ and maximum feed $f_{max}$.

$$f_{min} \leq f \leq f_{max} \tag{19}$$

(3) Cutting speed constraint.

$$\frac{\pi d_0 n_{min}}{1000} \leq v \leq \frac{\pi d_0 n_{max}}{1000} \tag{20}$$

where $v$ is the cutting speed; $d_0$ is the diameter of the workpiece to be machined, and $n_{min}$ and $n_{max}$ are the extreme values of the spindle speed.

(4) Surface roughness constraint. The surface roughness $R$ after machining should be less than the maximum allowable surface roughness $R_{max}$.

$$R \leq R_{max} \tag{21}$$

## 4. Case Study

A factory needs to process four holes in a given steel plate: material Q235, thickness 2.5 mm, production of 100,000 pieces. The existing equipment can be processed by turning, milling, and drilling. The product dimensions are shown in Figure 6.

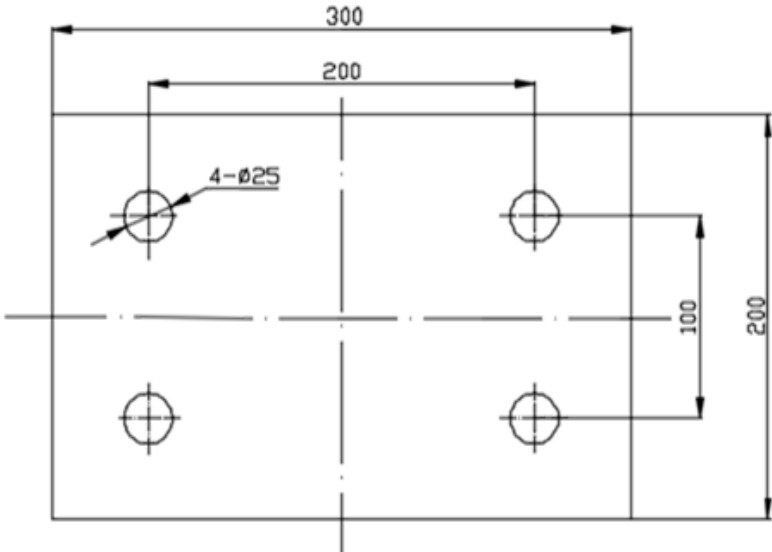

**Figure 6.** Product dimension.

### 4.1. Basic Situation Analysis of Existing Equipment Types

This product can be produced by three kinds of equipment in the enterprise. The basic parameters of the three kinds of equipment are expressed separately. The main parameters of the lathe are shown in Tables 1–3.

**Table 1.** Cutting parameters of the lathe.

| $v_{min}$ (m/min) | $n_{max}$ (m/min) | $f_{min}$ (mm/r) | $f_{max}$ (mm/r) | $a_{min}$ (mm) | $a_{max}$ (mm) |
|---|---|---|---|---|---|
| 45 | 120 | 0.15 | 0.75 | 2.5 | 7.6 |

**Table 2.** The tool life coefficients.

| $k_\gamma$ | $k'_\gamma$ | $\lambda_s$ | $r_\varepsilon$ |
|---|---|---|---|
| 75° | 4° | −5° | 1 mm |

**Table 3.** Optimizing model parameters.

| $K_{NFC}$ | $K_{NFC}$ | $K_{\gamma_0 FC}$ | $K_{\lambda_c FC}$ | $C_{FC}$ | $C_{FC}$ | $yF_C$ | $n_{F_c}$ | $\frac{1}{\alpha}$ | $\frac{1}{\beta}$ |
|---|---|---|---|---|---|---|---|---|---|
| (1.02, 3) | 0.92 | 1 | 1 | 2795 | 1 | 0.75 | (−0.1, 5) | 2.13 | 1 |

The tool-related parameters are shown in Table 2.

Tool life, cutting force coefficient, and other calculated correlation coefficients are shown in Table 3.

The milling machine is also one of the existing machines in the plant. The main parameters of the enterprise's milling machine are shown in Tables 4 and 5. NC milling machine specification parameters are shown in Table 4. Milling tool parameters are shown in Table 5.

**Table 4.** CNC milling machine specifications.

| n (r·min$^{-1}$) | p$_{max}$ (kW) | f$_z$ (mm·r$^{-1}$) | η | K$_m$ | M$_{max}$ (N·m) |
|---|---|---|---|---|---|
| 50~3500 | 2 | 0.02~5 | 0.8 | 0.2 | 20 |

**Table 5.** Milling tool parameters.

| Types of Knives | Knife Diameter(mm) | Number of Knife Teeth | Corner Radius r$_\varepsilon$ (/mm) |
|---|---|---|---|
| YT15 hard metal | 125 | 4 | 3 |

The drilling machine can also be used as processing equipment. The drilling machine is the MCV-810 Vertical Processing Center. Drilling machine specifications and drilling tool parameters are listed separately in Tables 6 and 7.

**Table 6.** Drilling machine specifications.

| Project | Unit | Number |
|---|---|---|
| X axis trip | mm | 810 |
| Y axis trip | mm | 510 |
| Z axis trip | mm | 560 |
| Workbench area | mm | 1000 × 510 |
| Main shaft speed | rpm | 8000 |
| Spindle motor specifications | kw | 15/10 |
| X/Y/Z axis fast speed | m/min | 15/15/12 |
| Maximum cutting speed | mm/min | 7000 |
| Machine tool power | kVA | 20 |
| System | FUNAC | series |

**Table 7.** Drilling tool parameters.

| Essential Parameter | Internal Circle Diameter | Thickness | Aperture | Horn R | Relief Angle |
|---|---|---|---|---|---|
| Number | 3.97 | 1.59 | 2.3 | <0.2 | 5° |

*4.2. Optimization Algorithm Selection and Parameter Setting*

There are many excellent algorithms to solve multi-objective problems in the engineering field, and the gray wolf algorithm is one of them [50–53]. The gray wolf algorithm was proposed by Mirjalili et al. in 2014, and was inspired by the natural predation behavior of gray wolf populations. Similarly to other swarm intelligence algorithms, it can be used to solve complex problems in different fields [54,55]. The gray wolf algorithm is a group intelligence optimization method based on the ecological habits of gray wolf predation. It uses wolves of different social classes to jointly guide the wolves to locate their targets and to realize the process of finding prey, surrounding prey, tracking prey, and capturing prey. The gray wolf algorithm has the characteristics of a simple structure, few parameters to be adjusted, and is easy to implement. In this paper, the update operator of the algorithm was redesigned, and the crossover mutation operation was added to solve the model. The gray wolf algorithm runs were as shown in Figure 7.

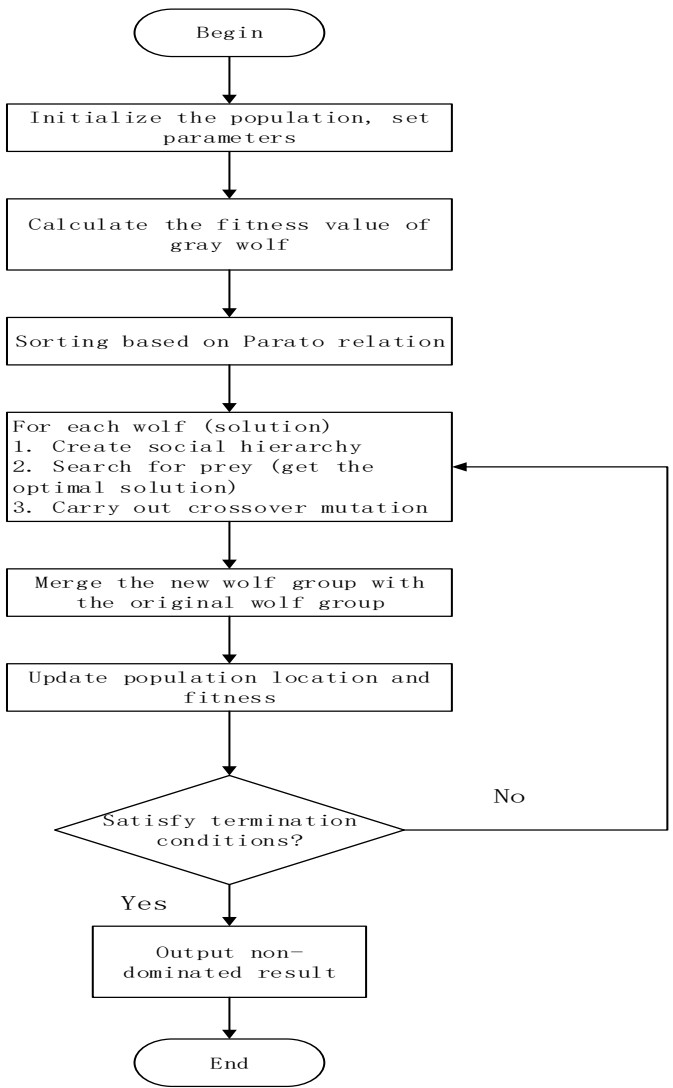

**Figure 7.** The Gray wolf algorithm flow chart.

The gray wolf algorithm was programmed by Matlab2014, the population number was 150, the maximum iteration number was 300, the crossover rate was 0.75, and the mutation rate was 0.2.

### 4.3. Optimization Results

The curves of optimal energy consumption and iteration times for different equipment can be obtained by the gray wolf algorithm, and are shown in Figures 8–10. The curves of optimal carbon emission and iteration times for different equipment can be obtained by the gray wolf algorithm, and are shown in Figures 11–13. Through simulation, the optimal energy consumption and the minimum carbon emissions can be obtained when the same product is processed with different equipment.

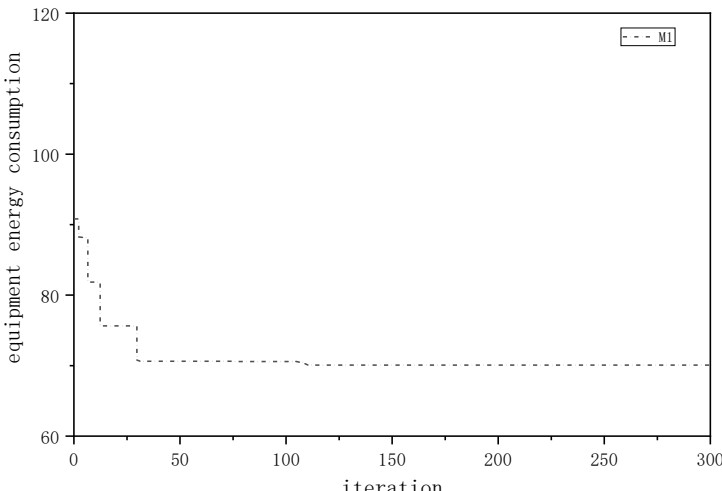

**Figure 8.** Energy consumption convergence curve of lathe.

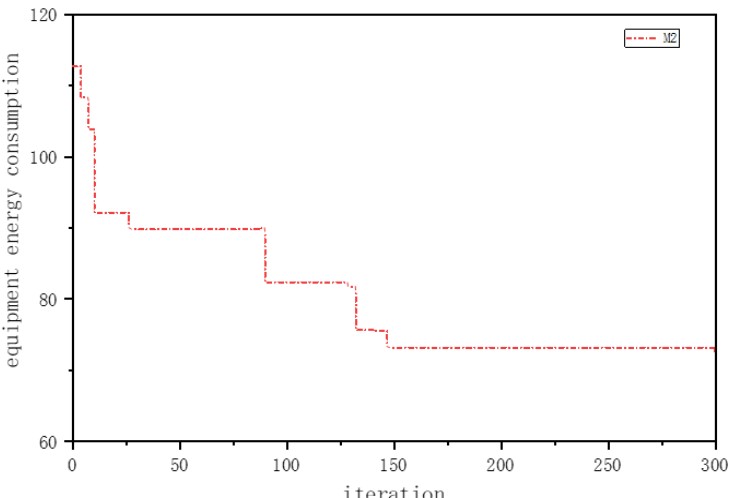

**Figure 9.** Energy consumption convergence curve of milling machine.

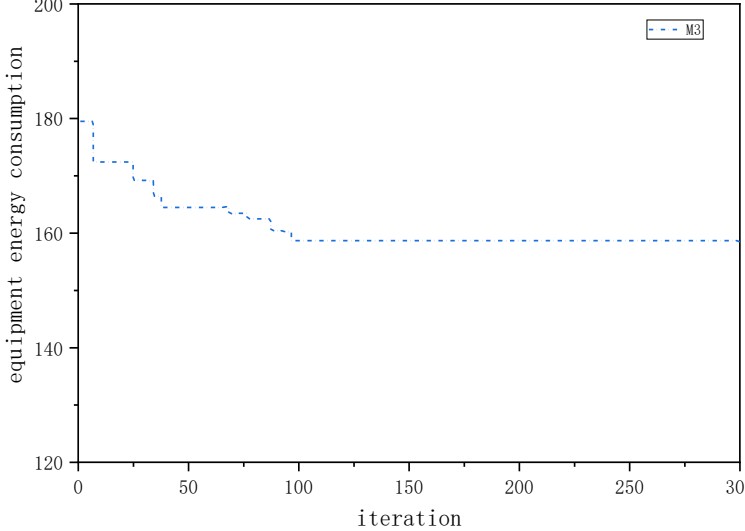

**Figure 10.** Energy consumption convergence curve of drilling machine.

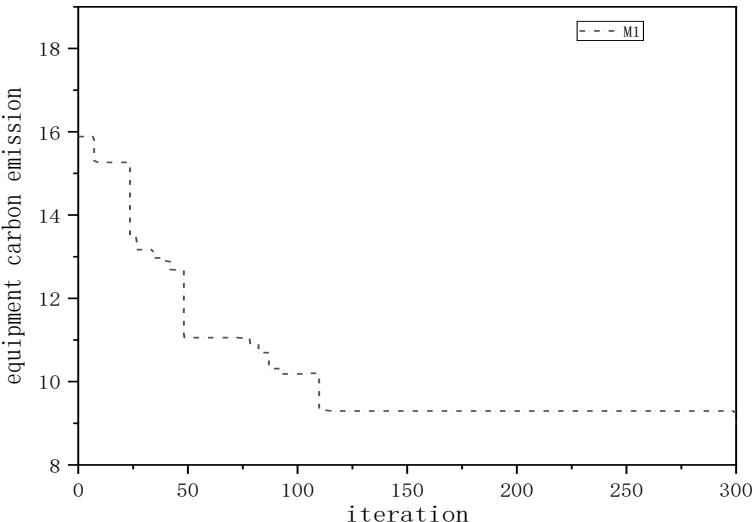

**Figure 11.** Curve of carbon emission convergence of lathe.

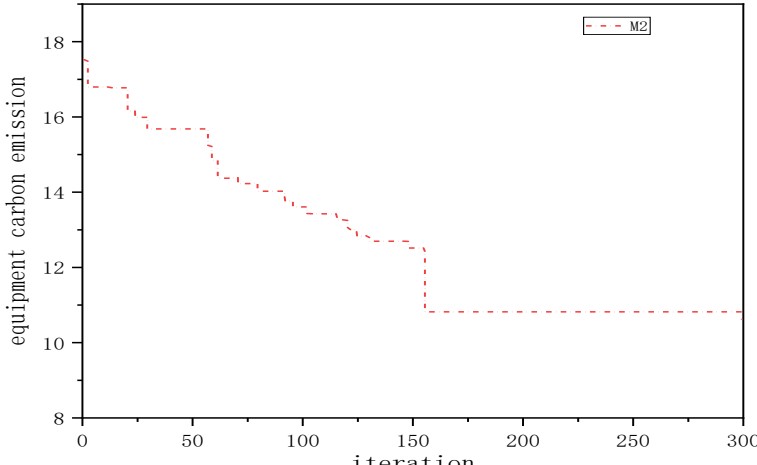

**Figure 12.** Carbon emission convergence curve of milling machine.

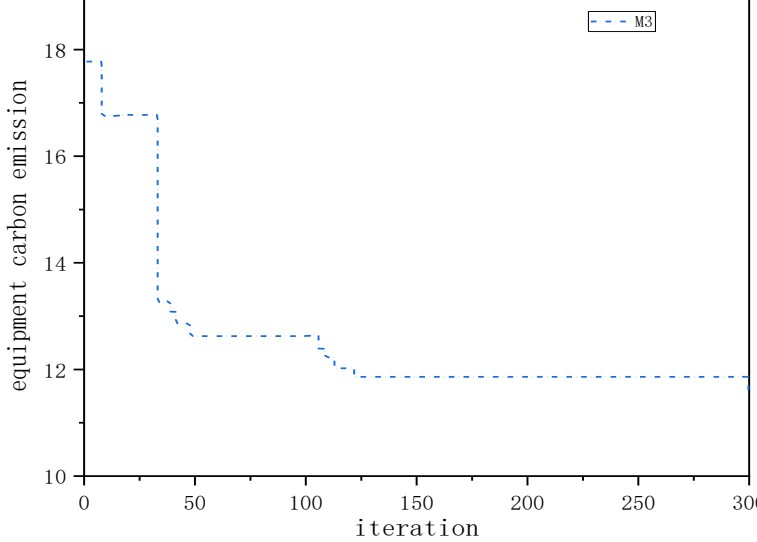

**Figure 13.** Convergence curve of drilling machine carbon emission.

*4.4. Analysis of Optimization Results and Discussion*

A multiple equipment unified energy consumption and carbon emission calculation model was proposed. The minimum energy consumption and minimum carbon emissions of the same product produced by different equipment were obtained by the grey wolf algorithm. Based on the optimal energy consumption and the minimum carbon emissions, the green levels of different equipment can be analyzed. Some problems observed in this study require further discussion.

4.4.1. Comparison of Different Equipment: Optimization Results

Table 8 shows the optimization results for the same product on different equipment. The ability of the model to analyze and evaluate the green development level of the same product on different equipment is evident. The first type of processing tool listed in Table 8 was undoubtedly the most energy-saving and lowest emitter of carbon. The second and third tools consumed 1.039 and 2.264 times as much energy as the first, respectively, and also emitted 1.182 and 1.29 times as much carbon, respectively. It is recommended that the first tool be used to process this product.

**Table 8.** Comparison of optimization results of different types of equipment.

| Equipment | Energy Consumption | Carbon Emissions |
|---|---|---|
| Lathe | 69.73 | 8.97 |
| Milling machine | 72.46 | 10.61 |
| Drilling machine | 157. 89 | 11.58 |

4.4.2. Comparative Benefits between the Proposed Method and Literature

In order to save energy and reduce emissions in the production process, some multi-objective optimization methods and algorithms have been reported at home and abroad. [7–26]. However, these studies are only static calculations of different types of production equipment, reporting qualitative analyses of multiple types of equipment. Because they only offer qualitative selection research, the data cannot provide a comprehensive basis for production enterprises to choose production equipment objectively. In this study, a unified energy consumption and carbon emission model for various types of equipment was established based on the dynamic characteristics of different equipment during operation. The model was solved by the grey wolf algorithm, and the quantitative result gave the optimal value of each equipment production process objective. The objective optimum values of different equipment production processes can provide quantitative comparison for enterprises to choose production equipment in line with the green development concept.

4.4.3. Practical Implications and Future Steps

A method for analyzing and evaluating the energy consumption and carbon emission level of the same product made by different equipment was proposed and verified through the production process of products to help quantify the selection of the best processing equipment. This study may be helpful to improve knowledge and understanding of the energy consumption and carbon emission levels of the same product processed by different equipment. It will also provide ideas for the government to promote green development. This study put forward, not only from the theoretical perspective, but also from the perspective of practical application, an analysis and evaluation method for ascertaining energy consumption and carbon emission levels of production with different equipment. However, this paper only considers the impact of the same product made with different kinds of equipment. Future research should include a comprehensive analysis and evaluation of the effects of enterprise personnel, equipment, raw materials, and other factors on the level of green development.

## 5. Conclusions

Analyzing and evaluating the green development level of product manufacturing processes is a complex problem that can not only affect the production decision-making of enterprises, but also promote the realization of socially sustainable green development. Based on the concept of green development, this paper establishes a set of green level analysis and evaluation methods for product manufacturing processes, including a unified-calculation energy consumption and carbon emission model for different kinds of equipment. This method can comprehensively consider the energy consumption and carbon emission of the same product produced by multiple kinds of equipment, and help enterprises to quickly select the appropriate equipment for production according to the demand.

1.  Based on the concept of green development, a set of methods for analyzing and evaluating energy consumption and carbon emissions in the product manufacturing process was established. This paper analyzes the influence of different factors on the green development level of a manufacturing process, and establishes the logical relationship between the selection of equipment and other factors.
2.  This paper established a unified calculation model of the energy consumption and carbon emission level of products made using different kinds of equipment. The model considers the characteristics of the operation of each tool and sets the specific parameters respectively.
3.  The grey wolf algorithm was used to optimize the model for calculating the energy consumption and carbon emissions of various equipment.

The results show that this method can analyze and evaluate the energy consumption and carbon neutralization level of the same product under different production processes, and provide suggestions for enterprises to enable them to choose through quantitative and qualitative analysis the production equipment suitable for supporting the concept of green development.

**Author Contributions:** Conceptualization, Y.X., R.Z., W.Y. and X.Z.; methodology, Y.X. and R.Z.; software, W.Y.; validation, R.Z.; formal analysis, Y.X.; visualization, R.Z.; supervision, X.Z. and W.Y.; project administration, W.Y. and Y.X.; funding acquisition, R.Z., W.Y. and X.Z. All authors have read and agreed to the published version of the manuscript.

**Funding:** The authors are grateful for the financial support for this research from the Key projects of the National Social Science Foundation: Research on ecological labor theory and its practice path oriented to green development (21AKS017); National Science Foundation, China (No.s 51975432, 61862051); China Education Department of Hunan Province (Project number: 21B0695); Project of Hunan social science achievement evaluation committee in 2022 (Project number: XSP22YBC081); the Science and Technology Foundation of Guizhou Province under Grant No. [2019]1299; the Top-notch Talent Program of Guizhou province under Grant No. KY[2018]080; the Program of Qiannan Normal University for Nationalities under Grant Nos. QNSY2018JS013, QNSYRC201715.

**Institutional Review Board Statement:** Not applicable.

**Informed Consent Statement:** Not applicable

**Data Availability Statement:** Not applicable.

**Conflicts of Interest:** The authors declare no conflict of interest.

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
