# Peer review of "Analysis and Evaluation of Energy Consumption and Carbon Emission Levels of Products Produced by Different Kinds of Equipment Based on Green Development Concept"

_sustainability, doi:10.3390/su14137631_

Round 1
Reviewer 1 Report
Even though the manuscript is promising, still needs revised. I have some comments and clarification quests that I list here below.
Chapter 1 and 2 needs revised due to the fact that the given information does not have coherence between them
Page 2 / line 95: “The production of industrial products is an important basis for the sustainable operation of society” please provide references
Paragraph 2.1 need references in order to support the given information
The paper must be check by a native English speaker.
Author Response
Chapter 1 and 2 needs revised due to the fact that the given information does not have coherence between them
Response: Thanks for your careful review and good advice. We have revised the paper based on this suggestion
Page 2 / line 95: “The production of industrial products is an important basis for the sustainable operation of society” please provide references
Response: Thanks for your careful review and good advice. We have provided references based on this suggestion
The paper must be check by a native English speaker.
Response: Thanks for your careful review and good advice. We have invited a native English speaker checked the paper.
Reviewer 2 Report
Here are my suggestions
1. Figure 1 is reported in the text but not explained in the text.
2. The equations` font sizes are not consistent
3. The concept of Green Development is wide; however, in this study, it’s not discussed fully discussed
4. The theoretical framework is missing. Why? Kindly incorporate it
5. Some tables are out of the size of the Sustainability template. This issue must be fixed. (Table 2)
Author Response
- Figure 1 is reported in the text but not explained in the text.
Response: The authors appreciate the helpful comment by the reviewer. We have revised the paper based on this suggestion.
- The equations` font sizes are not consistent
Response: The authors appreciate the helpful comment by the reviewer. We have we have revised the paper based on this suggestion.
- The concept of Green Development is wide; however, in this study, it’s not discussed fully discussed
Response: The authors appreciate the helpful comment by the reviewer. We have we have discussed concept of Green Development in introduction first paragraph based on this suggestion.
- The theoretical framework is missing. Why? Kindly incorporate it
Response: The authors appreciate the helpful comment by the reviewer. We have added the framework based on this suggestion.
- Some tables are out of the size of the Sustainability template. This issue must be fixed. (Table 2)
Response: The authors appreciate the helpful comment by the reviewer. We have we have fixed the tables size framework based on this suggestion.
Reviewer 3 Report
This manuscript addresses an interesting issue regarding the optimization of the energy consumption and carbon emissions associated with product manufacturing. This is an important topic in the international agenda towards a more sustainable and greener economy. A mathematical model is presented and applied to a specific case study. Three different machines are used and their output compared in terms of energy requirements and associated CO2 emissions. A comprehensive literature review is provided, as well as a discussion of the results. Finally, the list of references is adequate. These are the main strengths of the paper.
There are however several weaknesses in the manuscript, which I consider more important than the perceived strengths. My main concern is about the originality of the manuscript. I think it is important that the authors present the gap they found in the literature (as reviewed in refs. [7] to[25]) in order to support the research motivation included in lines 95 to 104. Reading section 4.4, I get the impression that an optimization was made in parallel for three different (and alternative) equipment. As such, what is the difference to other studies using the same methodology but for each equipment at a time? Please explain in more detail, for the sake of transparency and to better understand the contribution of the manuscript to the body of knowledge. A list of additional weaknesses that should be addressed to improve the overall quality of the manuscript follows:
1) The manuscript needs revision of the English language by a native speaker. Grammatical errors and inconsistencies throughout the manuscript detract from the easiness of reading;
2) If multi-equipment is on the basis of the originality of the manuscript, the title did not reflect this;
3) Line 42: the problem is not the “friendly interaction”, but the opposite;
4) Line 65: “[148]” should read “[14]”;
5) Figure 1 is not cited in the body text;
6) Line 137: “chart 1” should read “Figure 2”;
7) Chapter 3: For the sake of clarity, the way equations and their parameters are described should be revised;
8) Line 206: “KGCC)2” should read “kgCO2”;
9) Lines 208-209: please include a reference;
10) Subsection 3.3: Constraints should be further explained.
Author Response
I think it is important that the authors present the gap they found in the literature (as reviewed in refs. [7] to[25]) in order to support the research motivation included in lines 95 to 104.
Response: We really appreciate the reviewer’s helpful advice; we have revised the paper based on this suggestion
Reading section 4.4, I get the impression that an optimization was made in parallel for three different (and alternative) equipment. As such, what is the difference to other studies using the same methodology but for each equipment at a time? Please explain in more detail, for the sake of transparency and to better understand the contribution of the manuscript to the body of knowledge. A list of additional weaknesses that should be addressed to improve the overall quality of the manuscript follows:
Response: We really appreciate the reviewer’s helpful advice; we have revised the paper 4.4.2 based on this suggestion
1) The manuscript needs revision of the English language by a native speaker. Grammatical errors and inconsistencies throughout the manuscript detract from the easiness of reading;
Response: We really appreciate the reviewer’s helpful advice; we have revised the English language.
2) If multi-equipment is on the basis of the originality of the manuscript, the title did not reflect this;
Response: We really appreciate the reviewer’s helpful advice; we have revised the title based on this suggestion.
3) Line 42: the problem is not the “friendly interaction”, but the opposite;
Response: Thank you for your good suggestions. We have changed the word based on this suggestion.
4) Line 65: “[148]” should read “[14]”;
Response: We really appreciate the reviewer’s helpful advice; we have revised the paper based on this suggestion
5) Figure 1 is not cited in the body text;
Response: We really appreciate the reviewer’s helpful advice; we have cited Figure 1 in the body text.
6) Line 137: “chart 1” should read “Figure 2”;
Response: Thanks for your careful review and good advice; we have revised the paper based on this suggestion
7) Chapter 3: For the sake of clarity, the way equations and their parameters are described should be revised;
Response: We really appreciate the reviewer’s helpful advice; we have revised the equations and their parameters based on this suggestion
8) Line 206: “KGCC)2” should read “kgCO2”;
Response: We really appreciate the reviewer’s helpful advice; we have revised the paper based on this suggestion
9) Lines 208-209: please include a reference;
Response: We really appreciate the reviewer’s helpful advice; we have revised the paper based on this suggestion
10) Subsection 3.3: Constraints should be further explained.
Response: We really appreciate the reviewer’s helpful advice; we have further explained the constraints.
Round 2
Reviewer 1 Report
All the comments were taken into consideration
Reviewer 3 Report
The authors took into account the reviewers’ comments and the overall quality of the manuscript has improved.